# Variational Disentangled Cross-domain Knowledge Alignment for Multimodal Recommendation

## Abstract

Multimodal recommendation systems have been widely used in e-commerce and short video platforms. Due to the large differences in data volume and data distribution in different business scenarios, cross-domain recommendation is studied to improve the effect of target domain by using rich source domain data. Some studies use encoders to represent domain information and design knowledge alignment to achieve cross-domain knowledge transfer. However, simple information representation and alignment methods are easily affected by noisy information and lead to negative transfer problems. The distribution of features in different domains also has a large deviation, which affects the effective transfer of knowledge. Therefore, we propose a Variational Disentangled Cross-domain Knowledge Alignment Method (VDKA) for multimodal recommendation. Specifically, we propose a variational multimodal graph attention encoder, which consists of variational autoencoder and graph attention encoder. Variational encoder can learn domain sharing and domain specific representations under multimodal data utilization. Then we introduce variational optimization objectives and disentangled representation objectives to improve the accuracy of domain representation. Furthermore, in order to solve the problem of domain knowledge distribution drift, adversarial learning is designed to realize cross-domain knowledge alignment. We conducted comprehensive experiments on four real-world multimodal data sets, and the experimental results show that our proposed VDKA method outperforms other state-of-the-art models. Ablation experiments have verified the effectiveness of our various designs.

## 1 Introduction

With the increasing richness of image, audio and text information, the recommendation model based on multi-modal data has gradually achieved better results (Wang et al. (2023); Yu et al. (2022)). Compared with the single modal information, users are more likely to be attracted by the commodity display and function introduction in the video (Chen et al. (2022); Guo et al. (2022)). Since these platforms have rich scenarios, the data amount and data distribution of each business scenario are quite different. Therefore, cross-domain recommendation has been attached importance to the study of how to use rich source domain data to improve the effect of target domain with sparse data (Kang et al. (2019); Cao et al. (2022b)). Some methods mainly add multimodal data to the model as side information (Chen et al. (2019); Deldjoo et al. (2021)). These models use visual and text encoders to extract semantic features from images and text, and cross with attribute feature construction features for prediction. The main idea of cross-domain recommendation is to transfer the source domain knowledge with rich feature information to the target domain, so as to improve the accuracy of matching items with users in the target domain (Zhu et al. (2021; 2022)).

Although the existing cross-domain research has achieved some results, there are still several problems in the multi-modal cross-domain recommendation. First of all, the key problem of cross-domain recommendation is how to use the knowledge of source domain to improve the model effect of target domain. Many methods map feature information to a semantic space directly using encoders and transfer knowledge by feature crossing or feature alignment on the representation of two domains. This rough representation method is easy to be disturbed by a lot of noisy information.

Moreover, the knowledge contained in different domains may be contradictory, and simple information alignment is likely to cause negative transfer problems (Cao et al. (2022a); Zang et al. (2022)). Second, although some methods design disentangled representation networks to map domain features to the same semantic space, the distribution of knowledge in different domains is still biased. Knowledge transfer on biased data distribution will lead to biased model learning. Thirdly, the effective utilization of different modal data is an important issue in the multi-modal recommendation, and it is not accurate to only take modal data as side information.

Considering some of the key problems mentioned above, we propose a new solution. We propose a Variational Disentangled Cross-domain Knowledge Alignment Method (VDKA) for Multimodal Recommendation, which can effectively improve the multi-modal cross-domain recommendation effect. Specifically, we propose a variational multimodal graph attention encoder, which consists of variational autoencoder and graph attention encoder. Variational autoencoder is designed to learn domain-shared representations and domain-specific representations. Graph attention encoder is used to extract multi-modal feature information effectively. Then we introduce variational optimization objectives and disentangled representation objectives to improve the accuracy of domain-shared and domain-specific representations. Furthermore, in order to solve the problem of domain knowledge distribution drift, adversarial learning is designed to realize cross-domain knowledge alignment. Finally, we combine several optimization objectives as the loss function for model training. As a summary, the main contributions of this paper are as follows:

- We propose a variational multimodal graph attention encoder. Specifically, variational autoencoder is designed to learn domain-shared representations and domain-specific representations. Graph attention encoder is used to extract multi-modal feature information effectively.
- We propose a variational disentangled cross-domain knowledge alignment method (VDKA) for multimodal recommendation. VDKA uses variational encoders to extract domain-shared and domain-specific representations. The cross-domain knowledge alignment is further realized through adversarial learning.
- We conducted comprehensive experiments on four real-world multimodal data sets. Experimental results show that our proposed VDKA method exceeds other state-of-the-art models. The ablation experiments have also verified the effect of each module proposed by us. We will make our data sets and code public to contribute to community development.

## 2 RELATED WORK

As an important research direction in the field of recommendation, multimodal recommendation has been widely studied (Du et al. (2022); Yu et al. (2022); Han et al. (2022)). A lot of research work is exploring the extraction and utilization of multimodal data to help models improve performance (Xu et al. (2023); Yu et al. (2022); Han et al. (2022)). Some research methods add multimodal data as auxiliary features to the model and achieve results (Chen et al. (2019); Deldjoo et al. (2021)). VBPR (He & McAuley (2016)) incorporates visual features extracted from product images into matrix decomposition to reveal the visual dimensions that most influence people's behavior. Many work begin to use self-supervised comparative learning to solve problems such as data sparsity (Xie et al. (2022); Tao et al. (2022); Yu et al. (2021)). MMGCL (Yi et al. (2022)) designs two enhancement techniques, modal edge discard and modal mask, to generate multiple views of users and projects, and introduces a negative sampling technique to learn the correlation between modes. Many work using graph neural network to extract multi-modal information has achieved good results (Zhao & Wang (2021); Yu et al. (2022); Wei et al. (2020)). MMGCN (Wei et al. (2019)) constructs a user-item dichotomous graph on each mode and enriches the representation of each node with the topology and characteristics of its adjacent nodes. MGAT (Tao et al. (2020)) transmits information in a single graph, and uses the gated attention mechanism to identify the different importance scores of different patterns to user preferences.

Many researches are exploring the use of source domain data information to improve the prediction effect of target domain, so as to achieve effective cross-domain recommendation (Hu et al. (2018); Zhao et al. (2019); Sheng et al. (2021)). These studies focus on the extraction of domain information and the transfer of cross-domain knowledge. Some research work mainly uses encoders to learn domain representations and uses cross-transfer modules to achieve knowledge alignment (Wang et al.

(2021); Xu et al. (2021)). DDTCDR (Li & Tuzhilin (2020)) extends ConNet by learning a potential orthogonal projection function to migrate the similarity of users across domains. BiTGCF (Liu et al. (2020)) uses LightGCN (He et al. (2020)) as an encoder to aggregate interactive information for each domain, and further introduces a feature transfer layer to enhance the two basic graph encoders. Some research work focuses on the disentangled of domain representation. CDRIB (Cao et al. (2022b)) uses an information bottleneck perspective to obtain information shared between domains. DisenCDR (Cao et al. (2022a)) proposes two mutual information based disentangled regularizers to separate domain sharing and domain specific information. DDGHM (Zheng et al. (2022)) proposes dual dynamic graph modeling and mixed metric training to improve the cross-domain recommendation effect. UniCDR (Cao et al. (2023)) can model different CDR scenarios generically by passing domain shared information.

## 3 PRELIMINARIES

This paper mainly studies the cross-domain scenario of multimodal recommendation. Our proposed research method can be conveniently extended to multiple field scenarios. For simplicity, we focus on two domains that have a common set of users in this work. We let $D^A = (U^A, V^A, I^A)$ and $D^B = (U^B, V^B, I^A)$ represent the interaction data of domain A and domain B, respectively. $U$ denotes the shared user set in both domains, $V$ denotes the item set of each domain, and $I$ represents the interaction edge set in each domain. In addition, there are two binary matrices $Y^A \in \{0, 1\}^{|U| \times |V^A|}$ and $Y^B \in \{0, 1\}^{|U| \times |V^B|}$ representing the interaction matrices of domain A and B, respectively. $Y_{i,j}$ denotes whether a user $u_i$ has interacted with item $v_j$ in the edge set $I$. In addition, we focus on multimodal recommendation scenarios. In this paper, we mainly consider the use of visual and textual modal data. We represent the multimodal information as $x_m$, where $m \in M = \{v, t, a\}$. $v$ represents visual features, $t$ represents textual features, and $a$ represents acoustic features.

## 4 METHODOLOGY

In this section, we give a detailed introduction to the proposed VDKA method. The overall architecture and components of the VDKA are shown in Fig. 1.

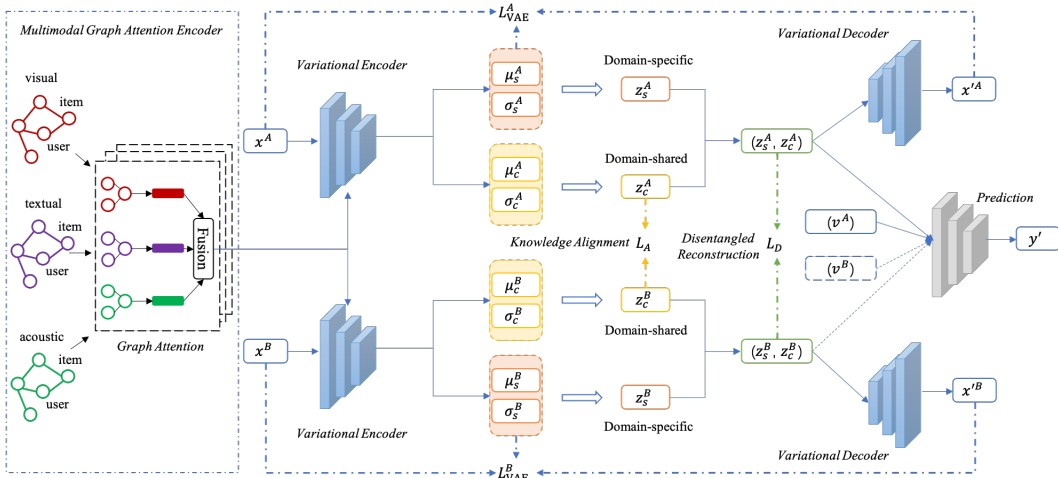

Figure 1: Overall architecture and components of the proposed VDKA method.

### 4.1 VARIATIONAL MULTIMODAL GRAPH ATTENTION ENCODER

#### 4.1.1 VARIATIONAL AUTO-ENCODER

Variational auto-encoder (VAE) is a combination of variational inference and auto-encoder, which is a kind of unsupervised generation model. VAE assumes that there exists an implicit variable $z$, and

the marginal distribution $P_\theta(x)$ can be calculated from $P_\theta(x) = \sum_z P_\theta(x, z)$. The joint distribution $P_\theta(x, z) = P_\theta(x|z)P(z)$ is satisfied between $z$ and input $x$. The distribution $P(z)$ of the implicit variable $z$ is assumed to satisfy a Gaussian distribution with a mean of zero and a variance of the unit vector. If an attempt is made to build a neural network approximating $P_\theta(x|z)$ without a suitable loss function, it will ignore $z$ and yield the trivial solution $P_\theta(x|z) = P_\theta(x)$. Therefore, this approach does not provide a good estimate of $P_\theta(x)$. The marginal distribution can be expressed in another form as $P_\theta(x) = \sum_x P_\theta(z|x)P(x)$. However, $P_\theta(z|x)$ is also difficult to solve. The goal of VAE is to find an estimable distribution that approximates $P_\theta(z|x)$, that is, an estimate of the conditional distribution of a latent variable $z$ given the input $x$. Therefore, based on variational inference and Bayes' theorem, the variational loss function can be obtained as follows:

$$L_{VAE} = E_{Q_\phi(z|x)}[log P_\theta(x|z)] - D_{KL}(Q_\phi(z|x)||P(z)) \leq log P_\theta(x) \tag{1}$$

The left side of the above equation is also called the evidence lower bound (ELBO). The inference model generates implicit variables $z$ from input $x$, and $Q_\phi(z|x)$ is similar to the encoder in the autoencoder model. $P_\theta(x|z)$ is similar to a decoder that extracts a sample from the inference model to reconstruct the input. To make the lower bound differentiable under the encoder parameter, the reparameterization trick is used to solve:

$$z = \mu_\phi(x) + \sigma_\phi(x) \odot \epsilon \quad where \quad \epsilon \sim N(0, 1) \tag{2}$$

where $\odot$ denotes the Hadamard product. $\mu$ and $\sigma$ represent the mean and variance of the multivariate Gaussian distribution corresponding to the latent variable $z$.

### 4.1.2 MULTIMODAL GRAPH ATTENTION ENCODER

In the multimodal scenario, users have different preferences for information of different modalities. However, simple modal representation cannot accurately describe the user's modal preference. We represent the interaction information as a bipartite user-item graph $G = \{(u, i)|u \in U, i \in I\}$, where $U$ denotes the user set and $I$ denotes the item set. We represent the set of modalities as $M \in \{v, t, a\}$, including visual features $v$, textual features $t$ and acoustic features $a$. Then, we design a multimodal graph attention encoder to learn representation vectors. The design details of the graph attention encoder are provided in Appendix D.

Based on the above graph attention method, we can obtain the representation matrix $H \in R^{|U| \times d}$ of multi-modal importance perception. Further, we bring $H^A$ belongs to domain $A$ into the variational autoencoder in Section 4.1.1 to generate the implicit variable Z as follows:

$$\mu^A = \phi(W_\mu^A H^A); \sigma^A = \phi(W_\sigma^A H^A); Z^A \sim N(\mu^A, [diag\{\sigma^A\}]^2) \tag{3}$$

where $W_\mu^A$ and $W_\sigma^A$ represent the parameter matrix. By performing variational graph attention encoding for data in both domains, the representations $Z^A$ and $Z^B$ can be obtained.

### 4.2 DOMAIN-SHARED AND DOMAIN-SPECIFIC REPRESENTATION LEARNING

Since the source domain and the target domain often have a big distribution difference, it is necessary to map the cross-domain information to the same semantic space to realize knowledge transfer. However, it is difficult to ensure that encoders can accurately represent cross-domain data in a semantic space. In addition, even if cross-domain data can be converted into lower-dimensional semantic space, the problem of domain distribution drift still exists. Therefore, in order to carry out cross-domain knowledge transfer accurately and effectively, we consider the disentangled learning of domain-shared representation and domain-specific representation. Domain-shared representation means common knowledge with the same semantic structure in multiple domains, and domain-specific representation means specific semantic knowledge unique only in a single specific domain. When domain representations with common semantic structure are extracted, domain knowledge transfer will be more effective to avoid negative transfer caused by useless cross-modal information.

### 4.2.1 VARIATIONAL ENCODING OBJECTIVE

Variational multimodal graph attention encoders provide a latent representation of multimodal perception. The latent variable $z$ can be inferred by an encoder with an approximate distribution of

$q_\phi(z|x)$. Based on the above analysis, we believe that there are modal shared implicit representations $z_c$ with common semantic structure and modal specific implicit representations $z_s$ with independent semantic structure on cross-domain data. For each variational encoder, there will be two branches to construct hidden variables $z_c$ and $z_s$ respectively. The joint probability distribution satisfied between variables is shown as follows:

$$P_\theta(x, z_c, z_s) = P_\theta(x|z_c, z_s)P(z_c)P(z_s) \tag{4}$$

where $P(z^c)$ and $P(z^s)$ represent prior distributions, which satisfy the Gaussian distribution with zero mean and unit variance. The distribution $P_\theta(x|z_c, z_s)$ can be seen as the generation distribution. Further, the distribution $q_\phi(z_c, z_s|x)$ can be converted to:

$$Q_\phi(z_c, z_s|x) = Q_\phi(z_c|x)Q_\phi(z_s|x) \tag{5}$$

where $Q_\phi(z_c|x)$ and $q_\phi(z_s|x)$ obey the Gaussian distribution, whose parameters are derived from the encoder output. Therefore, we rewrite VAE loss function in Section 4.2.1 to obtain the loss function based on latent variables $z_c$ and $z_s$ as follows:

$$L_V = E_{Q_\phi(z_c, z_s|x)}[log P_\theta(x|z_c, z_s)] - D_{KL}(Q_\phi(z_c|x)||P(z_c)) - D_{KL}(Q_\phi(z_s|x)||P(z_s)) \tag{6}$$

In particular, since we focus on the representation of multiple domains, we can obtain the loss function $L_V^A$ of domain A, and the loss function $L_V^B$ of domain B.

### 4.2.2 DISENTANGLED REPRESENTATION OBJECTIVE

In order to make full use of cross-domain information, we introduce information bottleneck theory to capture the correlation between domains. Specifically, we construct intra-domain and inter-domain information bottleneck regularizations from two different perspectives.

**Intra-domain information bottleneck regularization.** To ensure that the disentangled representation has the ability to recover the original feature information, we need to construct reconstruction targets for constraint. For instance, for the data $X^A$ of domain A, we can get the corresponding domain shared representation $Z_c^A$ and domain specific representation $Z_s^A$. We expect representation $Z_c^A$ and representation $Z_s^A$ to differ sufficiently to contain only domain-shared semantic information and domain-specific semantic information, respectively. We want $Z_c^A$ and $Z_s^A$ combined to have as little difference from $X^A$ as possible, meaning that the two representations can reconstruct the original feature information. Therefore, the regular loss of the intra-domain information bottleneck is defined as follows:

$$L_{intra} = \underbrace{I(Z_c^A; Z_s^A)}_{Minimality} - \underbrace{I(Z_c^A, Z_s^A; X^A)}_{Reconstruction} \tag{7}$$

where $I$ represent the mutual information operator.

**Inter-domain information bottleneck regularization.** For the shared representation $Z_c^A$ extracted from domain A, we believe that mutual information should be maximized with the shared representation $Z_c^B$ extracted from domain B. According to the shared representation $Z_c^B$ extracted from domain B and the specific representation $Z_s^A$ extracted from domain A, the original feature information $X^A$ of domain A should be reconstructed. Similarly, the original feature information $X^B$ of domain B should be reconstructed according to the shared representation $Z_c^A$ extracted from domain A and the specific representation $Z_s^B$ extracted from domain B. Introducing these reconstruction objectives enables the model to make full use of cross-modal information for knowledge transfer while learning accurate domain sharing and domain specific representations. Inter-modal information bottleneck regularization loss is defined as follows:

$$L_{inter} = \underbrace{-I(Z_c^A; Z_c^B)}_{Minimality} - \underbrace{I(Z_c^B, Z_s^A; X^A) - I(Z_c^A, Z_s^B; X^B)}_{Reconstruction} \tag{8}$$

We combine intra-domain and inter-domain information bottleneck losses together as the disentangled optimization loss $L_D$.

### 4.3 CROSS-DOMAIN KNOWLEDGE ALIGNMENT

There are great differences in data distribution and data sparsity among different domains, which makes the information in different fields have the problem of native semantic space heterogeneity. Therefore, we consider introducing adversarial learning to realize cross-domain learning. Instead of learning directly based on unconstrained implicit representation, we hope to reduce the distribution bias between $z_c^A$ and $z_c^B$. According to the covariate drift hypothesis, consistent optimization of source domain and target domain can make the predicted results of the two domains consistent. Motivated by (Long et al. (2018)), we introduce domain discriminator to optimize domain differences. One way to estimate the difference is to look at the loss of the domain classifier $G_d$, provided that the domain classifier's parameter $\theta_d$ has been trained to differentiate optimally between the two feature distributions. In order to reduce the distribution difference between $z_c^A$ and $z_c^B$, we seek the domain classifier parameter $\theta_d$ which minimizes the loss of the domain classifier. Optimization objectives based on domain discrimination can be expressed as:

$$E(\theta_f, \theta_y, \theta_d) = E(\theta_f, \theta_y, \hat{\theta}_d) + E(\hat{\theta}_f, \hat{\theta}_y, \theta_d) \tag{9}$$

$$E(\theta_f, \theta_y, \theta_d) = L_y(G_y(z_c^A, z_s^A; \theta_y)) - \lambda \sum_{i \in A}(G_d(z_{c,i}^A; \theta_d)) - \lambda \sum_{i \in B}(G_d(z_{c,i}^B; \theta_d)) \tag{10}$$

where $L_y$ represents the model prediction loss function. $z_{c,i}^A$ denotes the domain-shared representation of sample $i$ from domain A. $\hat{\theta}_f$ represents the optimized feature encoder parameter that generates $z_c$. $\hat{\theta}_f$ represents the optimized predcit network parameter used to minimize the loss of model prediction. $\hat{\theta}_d$ represents the optimized parameter of the discriminant network $G_d$, which is used to maximize the domain classification loss. Since the learning objective of the discriminator is opposite to that of the main task, gradient inversion layer (GRL) (Long et al. (2018)) is introduced to facilitate effective parameter updating. We let the optimization objective $E(\theta_f, \theta_y, \theta_d)$ as the knowledge alignment loss $L_A$.

### 4.4 OPTIMIZATION

In this paper, a variational multimodal graph attention encoder is introduced to extract multimodal information and construct domain-shared and domain-specific representations. The designed variational optimization objective can guide the model to learn accurate and effective feature representation. We denote the main loss of interaction prediction classification task as $L_y$. The final optimized loss function can be expressed as follows:

$$L = L_y + \alpha(L_V^A + L_V^B) + \gamma L_D - \lambda L_A \tag{11}$$

where $\alpha$, $\gamma$ and $\lambda$ represent hyperparameters that control the weight of different losses.

## 5 EXPERIMENTS

In this section, we conduct extensive experiments to answer the following questions:

- **RQ1** How does our VDKA model perform compared to the cross-domain and multi-modal state-of-the-art methods?
- **RQ2** How do different designs (e.g. variational encoding, disentangled representation and knowledge alignment) affect the performance of the VDKA model?
- **RQ3** How do the hyperparameter settings in the model affect the model performance?
- **RQ4** How does our approach contribute to cross-domain improvement?

### 5.1 EXPERIMENTAL SETUP

**Datasets.** We chose the Amazon dataset [1] as the experimental dataset. Amazon dataset (Lakkaraju et al. (2013)) is the real dataset extracted from e-commerce platform. Specifically, we select data

---

[1]`http://jmcauley.ucsd.edu/data/amazon/`

Table 1: Statistics of the four datasets.

| Cross-domain | Domain1-Items | Domain2-Items | Overlapped-Users | Train | Valid | Test |
|---|---|---|---|---|---|---|
| Movie & Book | 58,371 (Movie) | 104,895 (Book) | 12,746 | 118,779 | 23,756 | 15,838 |
| Food & Kitchen | 47,520 (Food) | 53,361 (Kitchen) | 8,575 | 50,845 | 10,169 | 6,779 |

sets in four domains from Amazon dataset as experimental data, including Movie, Book, Food and Kitchen. We pair data sets to construct cross-domain data sets "Movie & Book" and "Food & Kitchen". Text semantic representation is generated by Sentence-Bert. The statistical results are shown in Table 1.

**Evaluation metrics.** Following related work (Cao et al. (2022a)), we choose four widely used evaluation metrics, including Hit Rate (HR), Normalized Discounted Cumulative Gain (NDCG) and Mean Reciprocal Rank (MRR). We set the length of ranked list as 20. The implementation details are provided in Appendix B.

**Baseline methods.** To evaluate the performance, we compared the proposed VDKA model with three types of baselines: (1) CF-based methods, i.e., **BPRMF** (Rendle et al. (2012)) and **NGCF** (Wang et al. (2019)). (2) Multi-modal the state-of-the-art methods, i.e., **MMGCL** (Yi et al. (2022)), **LATTICE** (Zhang et al. (2021)) and **HCGCN** (Mu et al. (2022)). (3) Cross-domain the state-of-the-art methods, i.e., **DDTCDR** (Li & Tuzhilin (2020)), **BiTGCF** (Liu et al. (2020)), **DisenCDR** (Cao et al. (2022a)) and **DDGHM** (Zheng et al. (2022)). The details of baselines are provided in Appendix A.

## 5.2 PERFORMANCE COMPARISONS (RQ1)

We conducted a comprehensive experiment on two pairs of cross-domain datasets and compared our proposed VDKA method with other baseline methods. The experimental results are shown in Table 2. From the observation of the experimental results, we have the following findings. (a) Our proposed VDKA method outperforms all other SOTA cross-domain models and multi-modal models. The improvement is significant in all four experimental data sets. Experimental results verify the effectiveness of VDKA in multimodal cross-domain recommendation. (b) Compared with DDGHM and other cross-domain models, VDKA has significantly improved in the four data sets. The experimental results show that our method of disentangled representation and knowledge alignment is effective for cross-domain knowledge transfer. (c) Compared with cross-domain recommendation methods such as BiTGCF, HCGCN method focusing on multi-modal information utilization has better effect. This shows that the extraction and utilization of multi-modal features is as important as the cross-domain module in the multi-modal recommendation.

## 5.3 ABLATION STUDIES (RQ2)

In order to explore the influence of each module designed by us on the model effect, we conducted ablation experiments as follows. (a) We removed the variational multimodal graph attention encoder from the model and directly used two MLPS to learn domain-shared and domain-specific representations, denoted as *w/o VE*. (b) We removed the multimodal graph attention encoder, denoted as *w/o GA*. (c) We removed the disentangled optimization objective, denoted as *w/o DR*. (d) We removed the knowledge alignment objective, denoted as *w/o KA*. The experimental results are shown in Table 3. First, when we replaced the variational encoder with MLP, the model performance decreased significantly. This shows that the variational multimodal graph attention encoder is effective in extracting multimodal data and constructing domain representation. Secondly, when we removed the disentangled target and the knowledge alignment target respectively, the model effect decreases to a certain extent. The experimental results show that these two modules are important to the model.

## 5.4 SENSITIVITY ANALYSIS (RQ3)

To further explore the impact of these parameters on the model effect, we conducted sensitivity analysis on the following key parameters. (a) Impact of $\gamma$. We set the value of $\gamma$ to adjust from [0.2, 0.4, 0.6, 0.8, 1.0] to explore the impact on the model effect. The experimental performance on

Table 2: Overall performance comparison of all models on four data sets.

| | **Movie-domain** | | | **Book-domain** | | |
|---|---|---|---|---|---|---|
| | HR@20 | NDCG@20 | MRR@20 | HR@20 | NDCG@20 | MRR@20 |
| BPRMF | 0.1568 | 0.1041 | 0.0974 | 0.1407 | 0.0873 | 0.0837 |
| NGCF | 0.2319 | 0.1799 | 0.1713 | 0.2167 | 0.1619 | 0.1607 |
| MMGCL | 0.2658 | 0.2176 | 0.2114 | 0.2687 | 0.2019 | 0.1845 |
| LATTICE | 0.2911 | 0.2336 | 0.2227 | 0.2793 | 0.2201 | 0.2067 |
| HCGCN | 0.3253 | 0.2923 | 0.2767 | 0.3003 | 0.2649 | 0.2525 |
| DDTCDR | 0.2935 | 0.2621 | 0.2385 | 0.2568 | 0.2145 | 0.2019 |
| BiTGCF | 0.3134 | 0.2902 | 0.2775 | 0.2721 | 0.2407 | 0.2337 |
| DisenCDR | 0.3265 | 0.2944 | 0.2780 | 0.2867 | 0.2577 | 0.2398 |
| DDGHM | 0.3308 | 0.3011 | 0.2883 | 0.2981 | 0.2721 | 0.2635 |
| VDKA | **0.3625** | **0.3246** | **0.2932** | **0.3317** | **0.2892** | **0.2764** |
| Improvement | 5.25% | 4.78% | 5.81% | 6.16% | 6.77% | 6.77% |
| | **Food-domain** | | | **Kitchen-domain** | | |
| | HR@20 | NDCG@20 | MRR@20 | HR@20 | NDCG@20 | MRR@20 |
| BPRMF | 0.1383 | 0.0926 | 0.0888 | 0.1502 | 0.1178 | 0.0925 |
| NGCF | 0.2095 | 0.1914 | 0.1714 | 0.2253 | 0.2003 | 0.1848 |
| MMGCL | 0.2338 | 0.2085 | 0.1902 | 0.2539 | 0.2278 | 0.2049 |
| LATTICE | 0.2665 | 0.2407 | 0.2106 | 0.2802 | 0.2511 | 0.2306 |
| HCGCN | 0.2865 | 0.2611 | 0.2477 | 0.3047 | 0.2743 | 0.2639 |
| DDTCDR | 0.2396 | 0.2289 | 0.2019 | 0.2546 | 0.2352 | 0.2204 |
| BiTGCF | 0.2807 | 0.2545 | 0.2254 | 0.2937 | 0.2621 | 0.2532 |
| DisenCDR | 0.2992 | 0.2674 | 0.2558 | 0.3215 | 0.2967 | 0.2827 |
| DDGHM | 0.3023 | 0.2725 | 0.2537 | 0.3221 | 0.2942 | 0.2814 |
| VDKA | **0.3272** | **0.2936** | **0.2794** | **0.3438** | **0.3212** | **0.3035** |
| Improvement | 5.25% | 4.78% | 5.81% | 6.16% | 6.77% | 6.77% |

Table 3: Ablation experimental results. N denotes the NDCG metric.

| | **Movie** | | **Book** | | **Food** | | **Kitchen** | |
|---|---|---|---|---|---|---|---|---|
| | HR@20 | N@20 | HR@20 | N@20 | HR@20 | N@20 | HR@20 | N@20 |
| VDKA | **0.3625** | **0.3246** | **0.3317** | **0.2892** | **0.3272** | **0.2936** | **0.3438** | **0.3212** |
| w/o VE | 0.2915 | 0.2733 | 0.2553 | 0.2172 | 0.2633 | 0.2386 | 0.2763 | 0.2463 |
| w/o GA | 0.3304 | 0.3001 | 0.2943 | 0.2639 | 0.2906 | 0.2729 | 0.3186 | 0.2825 |
| w/o DR | 0.3344 | 0.3085 | 0.3082 | 0.2759 | 0.3186 | 0.2822 | 0.3325 | 0.3097 |
| w/o KA | 0.3368 | 0.3096 | 0.3124 | 0.2736 | 0.3153 | 0.2774 | 0.3268 | 0.3036 |

the four experimental data sets is shown in Fig. 2. We can see that with the increase of $\gamma$ value, the performance of the model on multiple data sets has an upward trend, but with the continuous increase of $\gamma$, there is basically no difference in the model effect. (b) Impact of $\lambda$. We let the parameter $\lambda$ adjust the value in [0.2, 0.4, 0.6, 0.8, 1.0]. As shown in Fig. 2, the performance of the model does not change significantly with the increase of $\lambda$ value. (c) Impact of embedding dimension $d$. The experimental results and analysis are provided in Appendix C.

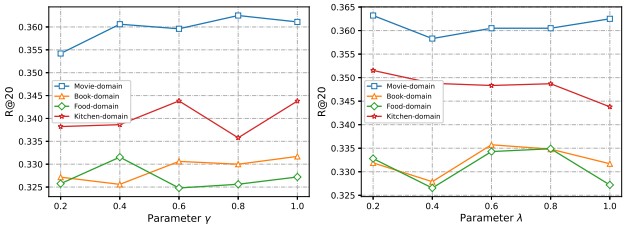

Figure 2: Sensitivity study of parameters $\gamma$ and $\lambda$.

Table 4: Cross-domain distribution discrepancy. The results are calculated from the $\mathcal{A}$-distance based on the cross-domain user representations.

| Model | Movie-domain & Book-domain | Food-domain & Kitchen-domain |
|---|---|---|
| DDTCDR | 1.3745 | 1.2491 |
| DisenCDR | 1.2529 | 1.1672 |
| DDGHM | 1.1957 | 1.1328 |
| VDKA-w/o DR | 1.1151 | 1.0469 |
| VDKA-w/o KA | 1.1072 | 1.0573 |
| VDKA | **1.0815** | **1.0151** |

## 5.5 DISTRIBUTION ANALYSIS (RQ4)

**Distribution Discrepancy.** According to domain adaptation theory (Ben-David et al. (2006); Liu et al. (2022)), a proxy $\mathcal{A}$-distance can be used to measure the difference between two different domains. The difference between the two domain distributions can be calculated by the formula $d_{\mathcal{A}}(S, T) = 2(1 - 2\epsilon(g))$, where $\epsilon(g)$ denotes the generalization error of a linear classifier $g$. The classifier $g$ is used to distinguish the source domain $S$ from the target domain $T$. The discrepancy in user domain distribution across the two cross-domain datasets are shown in Table 4. We can see that the distribution difference of VDKA proposed by us is significantly lower than that of DDGHM, resulting in better cross-domain performance. When we remove the uncoupling representation and the knowledge alignment, the distribution difference increases to a certain extent. This shows that our design method based on decoupling knowledge transfer can indeed improve the consistency of cross-domain distribution.

**Distribution Visualization.** To better show the difference in the distribution of the user embedding learned by different cross-domain methods, we use t-SNE representation for visualization. The visualization results are shown in Fig. 3. We can see that the difference in embedding distribution learned separately in the two domains is quite significant. This difference in distribution makes it difficult to transfer the knowledge of the model in the source domain to the target domain. Through the design of DDGHM model, the difference of embedding distribution can be alleviated to a certain extent. However, there is still a certain bias, and there may be a negative transfer problem. Our VDKA model solves the problem of distribution difference well and realizes accurate cross-domain representation learning through effective knowledge transfer. On the visualization of embedding, it can be seen that the method we propose indeed improves the consistency of cross-domain distribution, so as to provide stronger cross-domain recommendation capability.

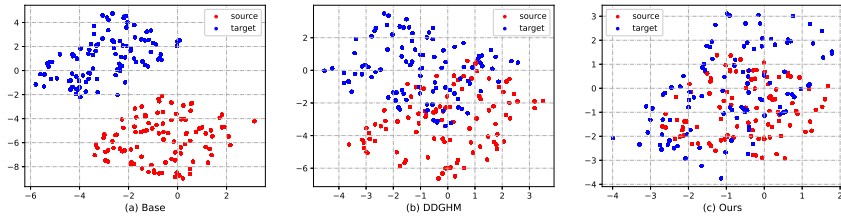

Figure 3: The t-SNE visualization of user embeddings on the Movie & Book cross-domain dataset.

## 6 CONCLUSION

In this paper, we propose a variational disentangled cross-domain knowledge alignment method for multimodal recommendation. Specifically, we propose a variational multimodal graph attention encoder that can effectively learn domain-sharing and domain-specific representations. Furthermore, adversarial learning is designed to realize cross-domain knowledge alignment. We conducted comprehensive experiments on multiple real-world data sets, and the experimental results show that our proposed VDKA method outperforms all other models. In the future, we will further explore the effective knowledge transfer method such as adversarial knowledge distillation.

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

## A    DETAILS OF THE BASELINES

The details of the baseline methods are as follows:

CF-based methods:

- **BPRMF** (Rendle et al. (2012)) mainly uses implicit feedback from users to sort items by the maximum posterior probability obtained from Bayesian analysis of problems.
- **NGCF** (Wang et al. (2019)) models the higher-order connection information in the embedding function and designs a neural network method to recursively propagate the embedding in the graph.

Multimodal-based methods:

- **MMGCL** (Yi et al. (2022)) carries out graph convolution on each modal bipartite graph and designs self-supervised optimization targets to improve the modal representation effect.
- **LATTICE** (Zhang et al. (2021)) designs a new modal-aware structure that learns the item structure of each modal and aggregates multiple modalities to obtain potential item diagrams.
- **HCGCN** (Mu et al. (2022)) designs graph convolutional networks and corresponding clustering losses to enhance user-item preference feedback and multimodal representation learning to make more accurate recommendations.

Cross-domain methods:

- **DDTCDR** (Li & Tuzhilin (2020)) develops a new potential orthogonal mapping to extract user preferences in multiple domains and constructs cross-domain dual knowledge transfer based on autoencoders.
- **BiTGCF** (Liu et al. (2020)) utilizes the high order connectivity of single domain user item graph through a feature propagation layer, and realizes the bidirectional transfer of cross-domain knowledge with common users as the bridge.
- **DisenCDR** (Cao et al. (2022a)) proposes a new domain-shared and domain-specific information decoupling method. Based on two decoupling regularizers with mutual information, DisenCDR can realize effective learning of sparse target domain.
- **DDGHM** (Zheng et al. (2022)) consists of dual dynamic graph modeling and mixed metric training. The former captures both intra-domain and inter-domain sequential transitions by constructing a two-level graph. The latter enhances user and item representation by adopting mixed metric learning.

## B    IMPLEMENTATION DETAILS

Following the data processing method adopted by (Zheng et al. (2022)), we select users who interact in both domains, and then filter users and items that interact less than 10 times. The visual features are provided by the data set and represented as 4096-dimensional embedding. Following (Mu et al. (2022)), we connect the item title, description and brand together to extract text features.

Following (Zheng et al. (2022)), We take the last item of the user interaction as a real sample of the prediction. We randomly selected 80% of the samples as the training set, 10% as the valiadation set, and 10% as the test set. For all baseline methods, we set the parameters according to the optimal overparameter mentioned in the original paper. For cross-domain recommendation methods that do not utilize multimodalities, we add uniformly extracted visual and textual representations to the model as side information. For fair comparison, the common hyperparameter settings for all methods are as follows. The implicit representation vector dimension is set to 64. Batch size is set to 1024 and the learning rate is set to 0.001. We use Adam (Kingma & Ba (2014)) as the training optimizer for all methods. The slope of LeakyReLU is set as 0.1. For our proposed VDKA method, the hyperparameters $\alpha$ and $L$ are set to 1.0. For simplicity, the hyperparameters $\gamma$ and $\lambda$ are set to 1.0. For all experiments, we repeated three times with different random seeds and reported average results.

## C SENSITIVITY ANALYSIS

Impact of embedding dimension $d$. We let parameter d adjust in [16,32,64,128,256,512]. The experimental results are shown in Fig. 4. We can see that the model's performance on all data sets gradually improves as the embedding dimension increases. As the dimension continues to increase, the performance of the model decreases. This is because the effect of the model is limited by the information of the data set, and too large vector dimension may lead to overfitting of the model.

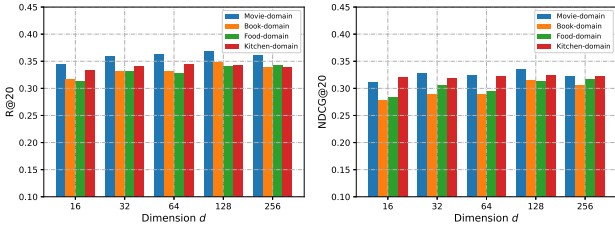

Figure 4: Effect study of embedding dimension $d$.

## D MULTIMODAL GRAPH ATTENTION ENCODER

For a node $e$ in a bipartite interaction graph, we use the aggregation function to calculate the information propagation of the neighbors $N_e = t|(e,t) \in E$, which is defined $h_m^{N_e} = f_{agg}(N_e)$. Considering the different importance of neighbors to the central node, we introduce attention network to control the different importance of information transmission from neighbors. The $f_{agg}$ is defined as follows:

$$f_{agg}(h_{m,t}^e) = LeakyReLU(\frac{1}{|N_e|} \sum_{t \in N_e} g_{att}(e,t) W_m^1 h_{m,t}^e) \tag{12}$$

where $h_{m,t}^e$ represents the node representation of item in modal $m$. $W_m$ denotes a learnable parameter matrix for aggregating valid information. Simply, we use a classic SENET (Huang et al. (2019)) attention network to calculate the importance of each neighbor, which is defined as follows:

$$g_{att} = g_{ex}(g_{sq}(h)) = \sigma_2(W_2 \sigma_1(W_1(\frac{1}{K} \sum_{k=1}^{K} h_{m,k}^e))) \tag{13}$$

where $h_{m,k}^e \in R^d$ denotes the $k$-th neighbor embedding. $W_1 \in R^{d \times \frac{d}{r}}$ and $W_2 \in R^{\frac{d}{r} \times d}$ represent the parameter matirx. As the relevant research analysis, id information plays an important role in the recommendation system. Therefore, the aggregate information can be obtained as follows:

$$h_{m,e} = LeakyReLU(W_m^3 h_{m,N_e} + LeakyReLU(W_m^2 h_m^e + h_{ID}^e)) \tag{14}$$

where $W_m^2$ and $W_m^3$ represent parameter matrix. $h_{ID}^e$ denotes the ID embedding of node $e$. Further, we can obtain the following new representations through the fusion of multiple modal representations that have explored the high-level connectivity information $h^e = \frac{1}{|M|} \sum_{m \in M} h_{L,m}^e$. Therefore, based on the above graph attention method, we can obtain the representation matrix $H \in R^{|U| \times d}$ of multi-modal importance perception.

