# OpenReview forum: "Multimodal Variational Disentangled  Knowledge Alignment for Cross-domain Recommendation"
_ICLR.cc/2024/Conference — Submitted to ICLR 2024_

### Official Review · Reviewer_AGJw · 2023-10-28

**Soundness:** 3 good
**Presentation:** 3 good
**Contribution:** 2 fair
**Rating:** 3
**Confidence:** 4

**Summary:**

This paper proposes a variational disentangled cross-domain knowledge alignment method for cross-domain recommendation. The domain-shared and domain-specific representations are learned through variational auto-encoders and adversarial training. Conducted experiments show the proposed method has better performance than compared baselines.

**Strengths:**

1. The paper is clearly written and easy to understand.
2. The proposed method sounds reasonable.
3. Experiments show clearly better results than baselines.

**Weaknesses:**

1. The motivation is weak, and the problems to tackle in this paper are not clear. Since the method mostly focuses on cross-domain recommendation, why must the authors use multimodal data?  There is almost no contribution for multimodal design.
2. The paper simply lists related works. There is no discussion on the differences between this work and literature, making its position unclear.
3. The designed methods are widely adopted in general domain adaptation, with small technical contributions.
4. The experiments are only conducted on two small datasets, hard to show the generality of the method. Also, there is no mean and variance or significance test.
5. The method uses multimodal information and cross-domain information, but the baselines only use one of them, which may be also unfair. It should be clearly stated the adopted information.

**Questions:**

Please see the weakness part.

---

> ### Comment · Reviewer_AGJw · 2023-11-22
> **Final comment**
>
> The authors did not address the mentioned limitations. I keep my original rating as a rejection.

---

### Official Review · Reviewer_XyWD · 2023-10-30

**Soundness:** 2 fair
**Presentation:** 2 fair
**Contribution:** 2 fair
**Rating:** 5
**Confidence:** 3

**Summary:**

This paper introduces variational multimodal graph attention encoder and variational disentangled cross-domain knowledge alignment method (VDKA). The graph attention encoder extracts node representations for each modality and then fuses them. The basic objective is to predict positive interaction of users and items.  The VAE encodes these user representations into domain-specific and domain-shared latent representations. The paper proposes intra-domain and inter-domain information bottleneck regularizations methods to get disentangled representations. The paper also proposes the cross domain knowledge alignment method to solve the problem of native semantic space heterogeneity of the two shared latent representations.

**Strengths:**

1. The paper nicely applied many methods to solve the multimodal multi-domain recommendation task and achieved best performance.
2. The paper conducted ablation experiments to verify each proposed method.
3. The paper shows low distribution discrepancy of VDKA compared to the previous models.

**Weaknesses:**

1. Applying the graph attention networks, multi-modal fusion, VAE, and disentangled representation is not novel.  I found similar disentangled representation methods  in https://arxiv.org/abs/2012.04251 and DisenCDR.
2. The explanation of the knowledge alignment method is unclear.

**Questions:**

1. What is the "covariate drift hypothesis"?
2. What is the definition of the output of G_d(z)?
3. Why is there no item (v) in the input of G_y in equation (10) as far as I understand that G_y is the binary classifier of positive interaction between user and item?
4. How to optimize the three parameters in equation (9) exactly?
5. In equation (11), why only L_A has minus coefficient? It is awkward that L_A is loss function and we want to maximize it. I'm so confused about whether we want to maximize equation (10) or not.
6. Are \lambda in equation (10) and (11) is same?
7. Does equation (10) mean that we want to minimize the loss of domain classifier while maximizing L_y? Why do we maximize the L_y?

---

### Official Review · Reviewer_UcLq · 2023-11-01

**Soundness:** 2 fair
**Presentation:** 2 fair
**Contribution:** 2 fair
**Rating:** 3
**Confidence:** 5

**Summary:**

This paper proposes a variational disentangled cross-domain knowledge alignment method(VDKA) for multimodal recommendation. Specifically,  a variational multimodal graph attention encoder learns domain-sharing and domain-specific representations, and adversarial learning is designed to realize cross-domain knowledge alignment. Experiments on multiple real-world datasets show VDKA  outperforms all baseline models selected by authors.

**Strengths:**

1. Concise language, easy to understand and read.
2. Cross-domain multimodal recommendation is a interesting and promising topic.
3. A novel and seemingly feasible solution is proposed and extensive experiments are conducted on two datasets.

**Weaknesses:**

1. The relationship between cross-domain recommendation and multimodal recommendation and the necessity of cross-domain multimodal recommendation are not clearly described.
2. The problems existing in the existing models described in the paper lack support and in-depth analysis.  There is a serious lack of literature research, and the baseline models are stuck at those published in 2022.
3. Poor organization of the paper. You should know that reviewers are not required to read the appendix. Excessive model details and insufficient description of the motivation for the proposed method and related methods. The placement of the table makes the part of experimental results inconvenient to read.
4. There are some errors in the paper, such as the calculation of performance improvement and the inconsistency between the fonts in the table and the fonts in the text.

**Questions:**

1.In the sensitivity analysis, the model performance has ups and downs instead of rising and then stopping or no obvious change as described in the paper. How to explain this phenomenon? In addition, when taking two extreme values (0.2 and 1.0), the performance difference on some datasets(such as lambda in book-domian) is very small. How to explain this?

---

### Meta-Review · Area_Chair_JV8j · 2023-12-12

**Metareview:**

This paper received originally the following negative ratings: 3, 5, and 3.
Poor organization of the paper, weak novelty, and insufficient experimental analysis are among the issues raised by the reviewers.
Rebuttal was not provided by the authors.
This paper is not acceptable for publication at ICLR 2024.

**Justification For Why Not Higher Score:**

N/A

**Justification For Why Not Lower Score:**

N/A

---

### Decision · Program_Chairs · 2024-01-16

Reject